

# Honey reduces the metastatic characteristics of prostate cancer cell lines by promoting a loss of adhesion

Sean D.A. Abel[1], Sumit Dadhwal[2], Allan B. Gamble[2] and Sarah K. Baird[1]

[1] Department of Pharmacology and Toxicology, University of Otago, Dunedin, New Zealand
[2] School of Pharmacy, University of Otago, Dunedin, New Zealand

## ABSTRACT

Honey has been shown to have a range of therapeutic effects in humans, with anti-inflammatory and anti-bacterial effects among those previously characterised. Here, we examine the possibility of New Zealand thyme, manuka and honeydew honeys, and their major sugar and phenolic components, reducing the development of metastatic cancer. Their activity was examined *in vitro*, in PC3 and DU145 prostate cancer cell lines, through measuring the compounds' effects on the metastatic characteristics of migration, invasion and adhesion. First, the phenolic compounds gallic acid, caffeic acid, quercetin, kaempferol and chrysin were quantified in the honeys using high performance liquid chromatography, and found in nanomolar concentrations. In a Boyden chamber-based migration assay, non-toxic concentrations of thyme and honeydew honeys reduced cell migration by 20%, and all phenolic compounds except caffeic acid also lowered migration, although a mixture of only the sugars found in honey had no effect. All of the honeys, phenolics and the sugar-only mixture reduced invasive movement of cells through extracellular matrix by up to 75%. Most notably, each of the three honeys and the sugar-only mixture reduced cell adhesion to collagen I by 90%. With the exception of quercetin, phenolic compounds did not reduce adhesion. Therefore, honey and its sugar and phenolic components can lower the metastatic properties of cancer cells, and may do this by preventing effective cell adhesion to the extracellular matrix. The sugars and phenol compounds of honey are much more effective in combination than individually.

## INTRODUCTION

Honey is made from nectar by honeybees (*Apis mellifera*), and has been shown to have potential therapeutic benefits in humans. Some of these characteristics, including anti-bacterial, anti-inflammatory, anti-oxidant and anti-hypertensive effects, have been demonstrated *in vivo* and some only *in vitro*. These properties are usually attributed to the honey's polyphenol components (*Alvarez-Suarez, Giampieri & Battino, 2013*).

The major constituents of honey are sugars, which make up more than 99% of the dry weight of honey (*White & Doner, 1980*). Enzymes, including invertase, diastase and glucose oxidase, in the honey stomach of the bees, convert the nectar-derived polysaccharides into

Corresponding author
Sarah K. Baird, s.baird@otago.ac.nz

monosaccharides (*Winston, 1991*). The sugar mixture consists mainly of fructose (40.5%), glucose (33.5%), maltose (7.5%) and sucrose (1.5%), which is consistent between honeys regardless of the origin of the nectar (*Cooper, Molan & Harding, 2002*). However, honey also has many other components, including amino acids, vitamins, minerals, polyphenols and enzymes (*White, 1978*). The polyphenols, including phenolic acids and flavonoids, make up most of this group. Their composition and proportions vary depending upon the source of honey, with five of the most prevalent and biologically active being quercetin, gallic acid, kaempferol, chrysin and caffeic acid (*Alvarez-Suarez, Giampieri & Battino, 2013*; *Kassim et al., 2010*; *Erejuwa, Sulaiman & Wahab, 2014*).

Honey has also been found to have a cytotoxic effect on cancer cell lines and in tumour-bearing animal models. It has been shown that honey can inhibit cancer cell proliferation and induce apoptosis in a range of cancers including breast, colon, liver and prostate cancers (*Fauzi, Norazmi & Yaacob, 2011*; *Hassan et al., 2012*; *Jubri et al., 2012*; *Tomasin & Gomes-Marcondes, 2011*; *Tsiapara et al., 2009*; *Wen et al., 2012*).

It is also likely that honeys may have an effect on metastasis, the formation of secondary tumours at other sites in the body, which is responsible for the majority of cancer deaths (*Mehlen & Puisieux, 2006*). In metastasis, cancer cells migrate away from the primary tumour site, invade through the local tissue and enter the circulation, infiltrate at a secondary site and finally re-establish tumour growth. The cell processes involved in metastasis are regulated by receptors such as integrins and intracellular signalling proteins that control cell adhesion to the surrounding extracellular matrix, degrade physical barriers such as the basement membrane via proteolysis and increase cell motility (*Liotta, Steeg & Stetler-Stevenson, 1991*). The cell moves as a result of cytoskeletal changes mediated by a cycle of actin polymerisation and depolymerisation controlled by the Rho family GTPases, which allows formation of membrane protrusions. The protrusions, and the cell body behind it, attach to and detach from the extracellular matrix in a regulated motion as the cell moves forward (*Pollard & Borisy, 2003*). The processes of migration, invasion and adhesion can each be measured separately *in vitro* to characterise the mechanism of action of effects on metastasis.

Two studies have investigated the effect of honey on metastasis using animal models. *Oršolić et al. (2005)* showed that 2 g/kg honey given orally daily before intravenous mammary or fibrosarcoma tumour cell injection into CBA mice reduced tumour formation in the lungs by around 70%, but had no effect if only given after tumour inoculation. An earlier study by the same group also examined colon adenocarcinoma in Y59 rats after intraveneous injection. In this study, it was found that honey, given orally for 10 days before tumour inoculation at 1 g/kg, reduced tumour nodules in the lungs by around 56% (*Oršolić et al., 2003*).

*In vitro*, any assessment of honey's potential anti-metastatic activity has been indirect. *Moskwa et al. (2014)* measured the activity of matrix metalloproteinase enzymes (MMPs), important for the invasion process, in U87MG glioblastoma cells, and found that they could be inhibited up to 95% by honeys, although the dose also caused high cytotoxicity. Otherwise, any related investigations have used only single polyphenols that are also found in honey. For example, gallic acid (3.5 $\mu$M) has been shown to reduce migration

in gastric cancer cells AGS by around 75% after 24 h in a scratch wound assay and 60% in a Boyden chamber assay after 48 h (*Ho et al., 2010*). Gallic acid also reduced invasion in the U87 glioma cell line by around 50% at 40 µg/mL (*Lu et al., 2010*). Caffeic acid (4 µg/mL) has been shown to reduce invasion in PC3 prostate cancer cells by around 50% (*Lansky et al., 2005*).

The aim of this study was to investigate whether whole honeys, or their main components, sugars or phenolic constituents, might have anti-metastatic properties. We first measured the concentrations of five major phenolic compounds, and then looked at the three main characteristics of metastatic cells, using *in vitro* assays to measure migration, invasion and adhesion.

# MATERIALS AND METHODS

## Materials

Dimethyl sulphoxide (DMSO) was from Scharlau (Barcelona, Spain). RPMI-1640 medium, foetal bovine serum (FBS), penicillin streptomycin (PS), PBS and trypsin were from Gibco (Carlsbad, CA). Bovine serum albumin (BSA), 3-(4,5-dimethylthiazol-2-yl)-2,5-diphenyl tetrazolium bromide (MTT), quercetin, caffeic acid, sucrose, maltose, fructose, D-(+)-glucose, formic acid, methanol, sodium bicarbonate, ethyl acetate, sodium hydroxide and hydrochloric acid were from Sigma-Aldrich (St. Louis, MO, USA). Thyme, manuka and honeydew honeys were kindly donated by New Zealand Honey Specialties Limited (Mosgiel, New Zealand). Kaempferol and chrysin were from Sapphire Biosciences (NSW, Australia). Gallic acid was from Abcam (Cambridge, UK). Fibronectin, collagen I and Matrigel® were from Corning (Corning, NY, USA).

## Preparation of honey and compounds

Thyme and manuka honey were from the Central Otago region of New Zealand, produced from the Thyme bush (*Thymus vulgaris)* or Manuka tree *(Leptospermum scoparium)*. Honeydew honey was collected in the Canterbury region of New Zealand, from the Beech Forest tree *(Nothofagus fusca)*. All honeys were produced by the European honey bee (*Apis mellifera)*. The sugar-only mixture, an artificial honey, was made by combining 40.5 g fructose, 33.5 g glucose, 7.5 g maltose and 1.5 g sucrose in 17 ml double distilled water (ddH$_2$O) (*Cooper, Molan & Harding, 2002*). Raw honeys were stored at room temperature, with minimal light exposure. Stock honey solutions were prepared by dissolving in warmed serum-free RPMI-1640 medium and sterilized using a 0.22 µm filter. Phenolic compounds were dissolved in 100% DMSO and filtered. Fresh preparations of honeys and compounds were made before each experiment.

## Preparation of phenolic extracts

The following method was adapted from *Wahdan (1998)*. Thyme, manuka and honeydew honeys were dissolved in warmed MilliQ water to give a final concentration of 20% (w/v) (10 mL or 15.48 g honey in 40 mL MilliQ). After extraction, dried samples were redissolved in 600 µL methanol.

*Free phenol extract preparation*

Each honey solution (50 mL) was adjusted to pH3.5 using concentrated HCl. Sodium bisulfite (1 g) and ethyl acetate (50 mL) were mixed with the honey solution in a separating funnel. The mixture was shaken for 1 min so that phenolic compounds moved into the organic phase. The honey solution was poured off, and the ethyl acetate layer transferred into a separate beaker. A further 50 mL ethyl acetate was added to the honey solution. These extraction steps were repeated six times. The ethyl acetate mixture (300 mL total) was concentrated using a rotary evaporator under vacuum ($\sim$240 mbar) at 30 °C. Dried compounds were reconstituted (methanol:ethyl acetate, 1:1), and dried under nitrogen to be stored at $-15$ °C.

*Total phenol extract preparation*

Extra preparation was required in order to hydrolyse phenolic compounds that were bound to sugars via ester bonds. Each honey solution (25 mL) was combined with NaOH (25 mL, 3N) and hydrolysed at room temperature for 4 h. Following this, pH was adjusted to 3.5 using concentrated HCl, and extraction carried out as for the free phenol extraction above.

## Quantification of phenolic compounds by high performance liquid chromatography (HPLC)

Phenolic separation was carried out using a CBM-20Alite Prominence HPLC (Shimadzu Corporation, Japan) on a reversed-phase Gemini 5 µm C18 110 Å, LC column (150 x 4.6 mm) (Phenomenex, USA). The method was adapted from *Kassim et al. (2010)*, and consisted of a gradient system to identify and quantify selected phenols in the extracted honey samples. The mobile phase was a binary solvent solution consisting of A = 0.25% formic acid and 2% methanol in water, and B = 100% methanol. The gradient method was as follows: 0 min 10% B, 20 min 40% B, 30 min 45% B, 50 min 60% B, 52 min 80% B, 60 min 90% B, 62 min 10% B until 65 min. Honey phenolic compounds were detected using a diode array where the spectra were monitored at 370 nm (kaempferol and quercetin), 325 nm (caffeic acid), and 270 nm (gallic acid and chrysin).

Calibration curves of standards were used to identify and quantify phenolic compounds within honey samples. Identification was completed by comparing standard retention time and absorbance spectrums against samples. Sample spiking was used to increase the confidence of peak identification. To resolve quantification issues due to split and shouldered peaks, the valley-to-valley integration method was employed.

## Cell culture

Human prostate cancer cell lines PC3 (more metastatic) and DU145 (less metastatic) were a gift from Prof. Rosengren (University of Otago, New Zealand). Cells were maintained in RPMI-1640 medium, supplemented with 5% FBS, 100 U/mL penicillin, 100 µg/mL streptomycin and 2 g/L sodium bicarbonate. Cells were incubated in 5% $CO_2$/95% $O_2$ humidified air at 37 °C.

## Boyden chamber migration assay

The undersides of the Boyden chamber membranes (8 µm pore size, 24 well plates, Transwells from Corning) were coated with collagen I (15 µL, 150 µg/mL 0.01M HCl in

PBS), left to dry under sterile conditions for 1 h, washed twice with PBS and dried for a further 1 h. $30 \times 10^3$ PC3 cells in serum free growth medium were seeded into the upper chamber. The bottom chamber contained 5% serum-containing growth medium. Both upper and lower chambers were treated for 48 h with compounds at previously established maximal non-cytotoxic concentrations as follows: 1% (w/v) honey, 25 µM quercetin, 10 µM gallic acid, 50 µM caffeic acid, 150 µM kaempferol or 100 µM chrysin (*Abel & Baird, 2018*). A vehicle control of 0.1% DMSO, the highest concentration added, was used. Following treatment incubation, MTT solution (0.5 mg/mL) was added to upper and lower chambers, and plates incubated for a further 3 h (*Mosmann, 1983*). Following this, a cotton tip was used to remove formazan crystals from the upper chamber. The undersides of the inserts were exposed to DMSO to dissolve formazan crystals from migrated cells. Absorbances were measured at 560 nm. Data were expressed as a percentage of migrated cells compared to a vehicle control.

## Boyden chamber invasion assay

This assay was used to determine the *in vitro* invasive ability of PC3 prostate cancer cells through Matrigel®, a solution containing extracellular matrix and basement membrane proteins resembling the structural parts of the tumour-surrounding stromal environment (*Hughes, Postovit & Lajoie, 2010*). The invasion of cells from the inside of the insert towards the underside was measured after the cells were exposed to compounds at previously established non-cytotoxic concentrations for 72 h: 0.5% (w/v) honey, 5 µM quercetin, 10 µM gallic acid, 50 µM caffeic acid, 150 µM kaempferol or 100 µM chrysin (*Abel & Baird, 2018*). A vehicle control of 0.1% DMSO, the highest concentration added, was used. Matrigel® was added to the upper chamber of inserts, and after 1 h, was washed with serum-free growth medium. The remainder of the method proceeded as for the migration assay, although the treatment duration was 72 h. Data were expressed as a percentage of migrated cells compared to a vehicle control.

## Cell adhesion assay

In order to assess the ability of honey to affect PC3 prostate cancer cell adherence *in vitro*, an assay to measure attachment was used (*Jin et al., 2000*). 96-well plates were coated with collagen I (20 µL, 5 g/mL, 0.01 M HCl in PBS) or fibronectin (20 µL, 5 µg/mL in PBS) and dried for 1 h. Plates were washed twice in PBS. BSA (1%, in PBS) was added to each well for 1 h. Wells were washed twice with PBS and left to dry for a further 2 h. $1 \times 10^4$ PC3 or DU145 cells in growth medium were added to each well. Cells were treated with honey or compound at concentrations that were non-toxic over 24 h as shown previously[24] and were left to incubate for 30 and 60 min (PC3) or 30, 60 and 90 min (DU145). The concentrations used were 0.5–5% for honeys, up to 150 µM for quercetin and kaempferol, up to 50 µM for chrysin and caffeic acid, and up to 10 µM for gallic acid. A vehicle control of 0.1% DMSO, the highest concentration added, was used. Wells were aspirated and washed twice in PBS. RPMI 1640 with 5% FBS with MTT (0.5 mg/mL) was added to each well and incubated for 3 h. The formazan crystals were solubilised in DMSO and absorbances were measured at 560 nm. Data were expressed as a percentage of vehicle control adhered cells.

## Statistics

Each experiment was set up in triplicate, and repeated three times. Results were expressed as mean ± SEM using GraphPad Prism 5 (GraphPad, San Diego, CA, USA). Data were considered significant at $P < 0.05$ (*). For cytotoxicity experiments, a two-way ANOVA followed by a Bonferroni post-hoc test was used to determine the effect of both concentration and time on cell death. Where normality was assumed, outliers were identified using the Online GraphPad Prism Grubbs' Test, and were removed from data sets according to recommendations based on calculated critical Z values.

# RESULTS

## Quantification of phenolic compounds in honey by HPLC

Since we were interested in quantifying the effect of whole honeys and their main phenolic constituents on the pro-metastatic properties of prostate cancer cell lines, we first optimised a method to extract and measure both total and free phenols, including gallic acid, caffeic acid, quercetin, kaempferol and chrysin, from the thyme, manuka and honeydew honeys. An HPLC gradient system was developed to ensure sufficient separation of the compounds, which were identified using calibration curves with standards, verified by sample spiking. The valley-to-valley integration method was used for analysis, due to frequent split and shouldered peaks (Fig. 1).

Table 1 shows the concentrations of free and total phenols found in the thyme, manuka and honeydew honeys, as both µg/100 g of honey and in nM, allowing comparison with the concentrations used in later experiments. The retention times, in minutes ($\pm$SEM, $n = 18$), were 4.3 ± 0.02 for gallic acid, 20.3 ± 0.02 for caffeic acid, 35.3 ± 0.12 for quercetin, 41.1 ± 0.1 for kaempferol and 50.0 ± 0.07 for chrysin.

The amounts of each phenol varied widely by up to tenfold between the three honey types. Gallic acid recorded the highest concentration among the phenolic compounds throughout all honeys with a total of 1,082.88 ± 16.25 µg/100 g free-phenol extracted honeydew honey, which becomes a low µM concentration in the 5% (w/v) honey solution used in later assays. Quercetin recorded the lowest concentration among all phenolic compounds, being only detectable in free phenol-extracted thyme and manuka honey. A higher recovery of gallic acid and caffeic acid were observed in most total phenol extracts, which were in contrast to all other compounds that showed a higher recovery in free phenol extracts.

## Effect of honey and its constituents on prostate cancer cell migration and invasion

Migration and invasion were tested in Boyden chambers, with the addition of Matrigel for the invasion assay. For these experiments, only the PC3 cell line was used, since the DU145 cell line is not very migratory, and was found not to move through the Boyden chamber. The tests were conducted using the previously reported highest non-toxic concentrations of honey of 1% for the 48 hour-long migration assay, and 0.5% for the 72 hour-long invasion assay (Abel & Baird, 2018).

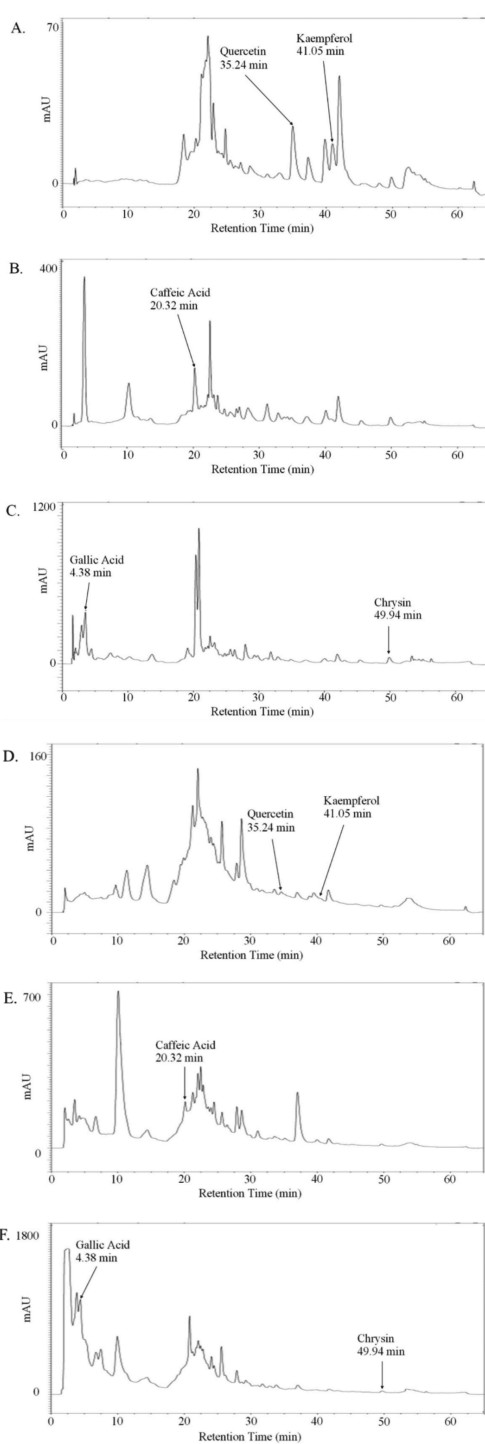

**Figure 1 Absorption chromatogram of phenolic compounds in thyme honey using HPLC.** Representative chromatogram of free (A–C) or total (D–F) phenols extracted from thyme honey using ethyl acetate and eluted through a C18 column using HPLC. (A,D) Detectors were set to 370 nm for quercetin and kaempferol, (B,E) 325 nm for caffeic acid and (C,F) 270 nm for gallic acid and chrysin. Peak retention times were compared to standards to identify compounds.

**Table 1  HPLC analysis of ethyl acetate extracted phenolic compounds from honey.**

| Compound | Honey | [μg/100 g honey] | | [5% w/v solution] (nM) | |
| --- | --- | --- | --- | --- | --- |
| | | Free | Total | Free | Total |
| Gallic Acid | Thyme | 94.2 ± 2.5 | 686.0 ± 15.2 | 276.8 ± 7.5 | 2016.0 ± 44.6 |
| | Manuka | 741.5 ± 27.4 | 584.5 ± 25.9 | 2179.0 ± 80.4 | 1718.0 ± 76.0 |
| | Honeydew | 48.1 ± 6.2 | 1082.9 ± 16.3 | 141.5 ± 18.2 | 3183.0 ± 47.8 |
| Caffeic Acid | Thyme | 90.2 ± 2.7 | 83.7 ± 2.7 | 250.4 ± 7.6 | 232.2 ± 7.5 |
| | Manuka | 44.8 ± 3.5 | 131.3 ± 17.1 | 124.3 ± 9.8 | 364.4 ± 47.3 |
| | Honeydew | 28.9 ± 2.5 | 96.6 ± 9.0 | 80.2 ± 7.0 | 268.0 ± 25.1 |
| Quercetin | Thyme | 38.2 ± 3.2 | – | 63.2 ± 5.3 | – |
| | Manuka | 10.6 ± 7.4 | – | 17.5 ± 12.2 | – |
| | Honeydew | – | – | – | – |
| Kaempferol | Thyme | 45.3 ± 13.8 | 33.1 ± 12.2 | 79.2 ± 24.1 | 57.9 ± 0.6 |
| | Manuka | 9.3 ± 2.6 | 12.2 ± 2.0 | 16.2 ± 4.6 | 21.3 ± 3.5 |
| | Honeydew | 11.2 ± 1.8 | 8.6 ± 0.2 | 19.6 ± 3.1 | 15.0 ± 0.3 |
| Chrysin | Thyme | 53.0 ± 0.3 | 38.1 ± 0.4 | 104.3 ± 0.5 | 75.0 ± 0.7 |
| | Manuka | 77.6 ± 0.8 | 72.1 ± 1.3 | 152.6 ± 1.7 | 141.9 ± 2.5 |
| | Honeydew | 46.4 ± 0.5 | 26.8 ± 0.6 | 91.2 ± 0.9 | 52.8 ± 1.1 |

**Notes.**

Compound concentrations (μg/100g honey) were calculated by comparing the area under the peak to standard calibration curves and were expressed as mean ± S.E.M ($n = 6$).
Concentrations of each compound were also calculated from within a 5% (w/v) solution of honey used throughout the study (nM).

The 1% thyme or honeydew honeys caused a small but significant reduction in PC3 cell migration compared to vehicle-treated control of 20.59 ± 4.90% and 18.90 ± 1.62% respectively (Fig. 2A). The manuka honey and the sugar-only mixture did not cause any reduction in cell migration, suggesting that non-sugar related compounds in honey may be responsible for the effect on migration.

All the non-toxic concentrations of phenolic compounds tested, with the exception of caffeic acid, caused a decrease in PC3 cell migration (Fig. 2B). Gallic acid (10 μM) caused the greatest reduction in migration of 49.10 ± 8.33% compared to vehicle (DMSO)-treated control. Together, these results demonstrated that phenolic compounds found in honey possess anti-metastatic activity against PC3 cells, however not all compounds have similar potency.

When invasion was measured, which looks at PC3 cell movement as well as the capacity to move through the Matrigel matrix, 0.5% (w/v) thyme, manuka or honeydew honey caused a significant reduction compared to control of 50.53 ± 4.10%, 60.44 ± 9.71% and 75.32 ± 0.19% respectively (Fig. 2C). The sugar-only mixture caused a 46.77 ± 17.01% reduction in cell invasion, however it was not statistically significant. These changes were much greater than the reduction of migration, suggesting that honey particularly reduces the cells' ability to move through Matrigel.

Invasion was also lowered by most of the phenolic compounds, with quercetin (10 μM) and caffeic acid (50 μM) resulting in reductions of 45.28 ± 0.64% and 14.44 ± 1.17% respectively (Fig. 2D). Further, chrysin (100 μM) and gallic acid (10 μM) reduced invasion by 39.57 ± 14.57% and 29.03 ± 15.86% respectively, however these were not statistically

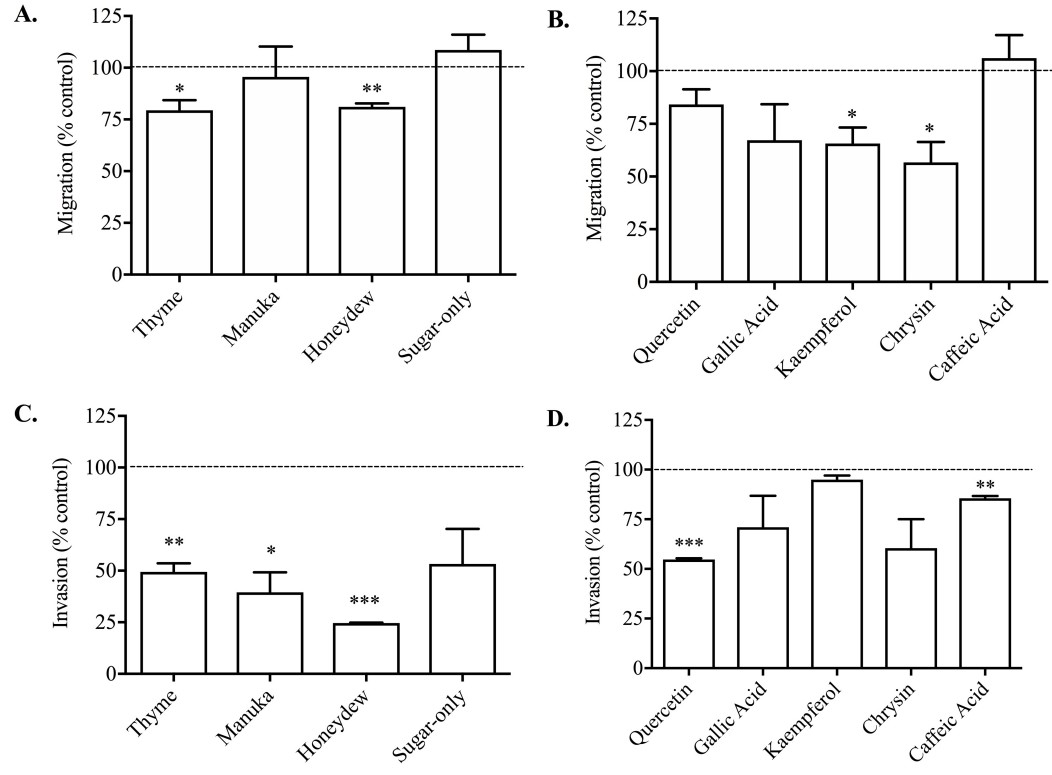

**Figure 2** **Effect of honey and honey-derived phenolic compounds on PC3 cell migration and invasion.** PC3 cells were placed in Boyden chambers and treated with 1% (w/v) thyme, manuka, honeydew or sugar-only mixture (A) or the phenolics quercetin (10 μM), gallic acid (10 μM), kaempferol (150 μM), chrysin (100 μM) or caffeic acid (50 μM) (B) for 48 h to measure migration or in Boyden chambers containing Matrigel for 72 h with 0.5 % (w/v) thyme, manuka, honeydew or sugar-only mixture (C) or the phenolics quercetin (10 μM), gallic acid (10 μM), kaempferol (150 μM), chrysin (100 μM) or caffeic acid (50 μM) (D) for 72 h to measure invasion. Experiments were completed in triplicate, and values were expressed as mean cell migration ± S.E.M ($n = 2$ or 3). Data were analysed using a two-tailed Student's $t$-test. * ($p < 0.05$) represents a significant difference between individual treatment and the vehicle- untreated cell control of 0.1% DMSO for phenolics and RPMI-only for honeys.

significant. Finally, kaempferol only recorded a reduction in invasion of 5.04 ± 2.11%. Overall, the reductions in invasion caused by the phenolic compounds were comparable to those in migration, with the exception of kaempferol and caffeic acid.

## Effects of honey and honey constituents on prostate cancer cell adhesion

Thyme, manuka and honeydew honeys caused a much greater, concentration-dependent decrease in cell adhesion, compared to their effects on migration and invasion, in both PC3 and DU145 cells ($p < 0.05$) (Fig. 3). This was assessed after 30 and 60 min of exposure to the honeys in PC3 cells, and 60 and 90 min in the DU145 cells, which required longer to become adherent. DU145 cells were much more sensitive to the effect of the honeys, with their adhesion to collagen I being nearly completely prevented by 2% (w/v) honey, down to 8.98 ± 4.65% of vehicle-treated control with honeydew honey after 60 min, whereas

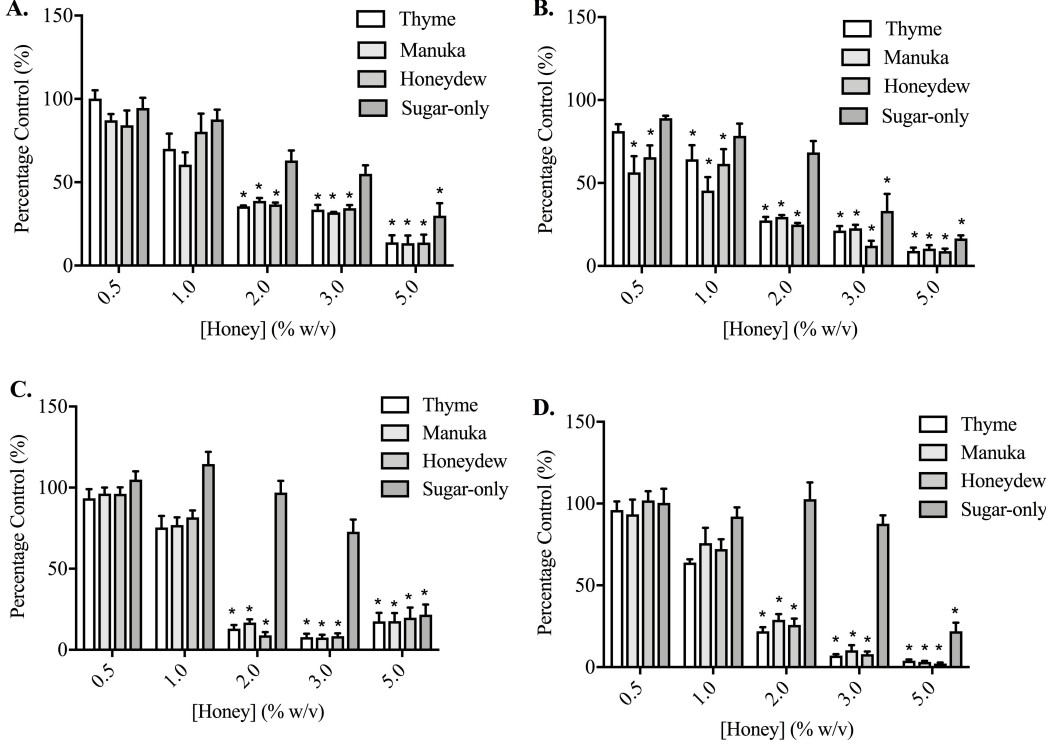

**Figure 3  Effect of honey on PC3 and DU145 cell adhesion to collagen I.** (A,B) PC3 and (C,D) DU145 cells were left to adhere to plates coated with collagen I for (A) 30, (B,C) 60 or (D) 90 min. Experiments were completed in triplicate, with results expressed as mean percentage of control ± S.E.M ($n$ = minimum of 3). Individual data were analysed using a two-way ANOVA followed by a Bonferroni post-hoc test, where $p < 0.05$ was required for a statistically significant difference. * represents a significant difference between the vehicle-treated control (0% w/v RPMI only) and individual treatment.

the PC3 cells required 5% (w/v), reaching 8.93 ± 3.69% with honeydew honey at 60 min. In PC3 cells, the sugar-only mixture was also effective at reducing adhesion, at 3% (w/v) (33.18 ± 25.05%) and 5% (w/v) (16.51 ± 4.68%) at 60 min, but in the DU145 cells, lower concentrations of the sugar-only mixture actually increased adhesion (although not significantly) before strongly lowering it at 5% (w/v), to 21.88 ± 13.01% at 90 min.

When cells were adhering to fibronectin, the results were very different (Fig. 4). In PC3 cells, while the honeys did lower adhesion in a statistically significant way, the effect was no longer large and nor was it concentration-dependent. The sugar-only mixture and the honeys gave similar levels of adhesion reduction, with the maximum reduction being found at 60 min with 4% (w/v) manuka honey, at 67.15 ± 8.33%. In DU145 cells, the effects were also much smaller than they had been with collagen I, although the reduction was much greater than with the PC3 cells. Adhesion was lowered to 42.33 ± 1.05% at 90 min by 3% (w/v) honeydew honey. Again, the effects were not concentration-dependent. With the DU145 cells, however, the changes in adhesion with the sugar-only mixture were very different from what was seen with the PC3 cells. With most concentrations, an increase in adhesion was found, up to 190.97 ± 5.46% with 4% (w/v) at 90 min.

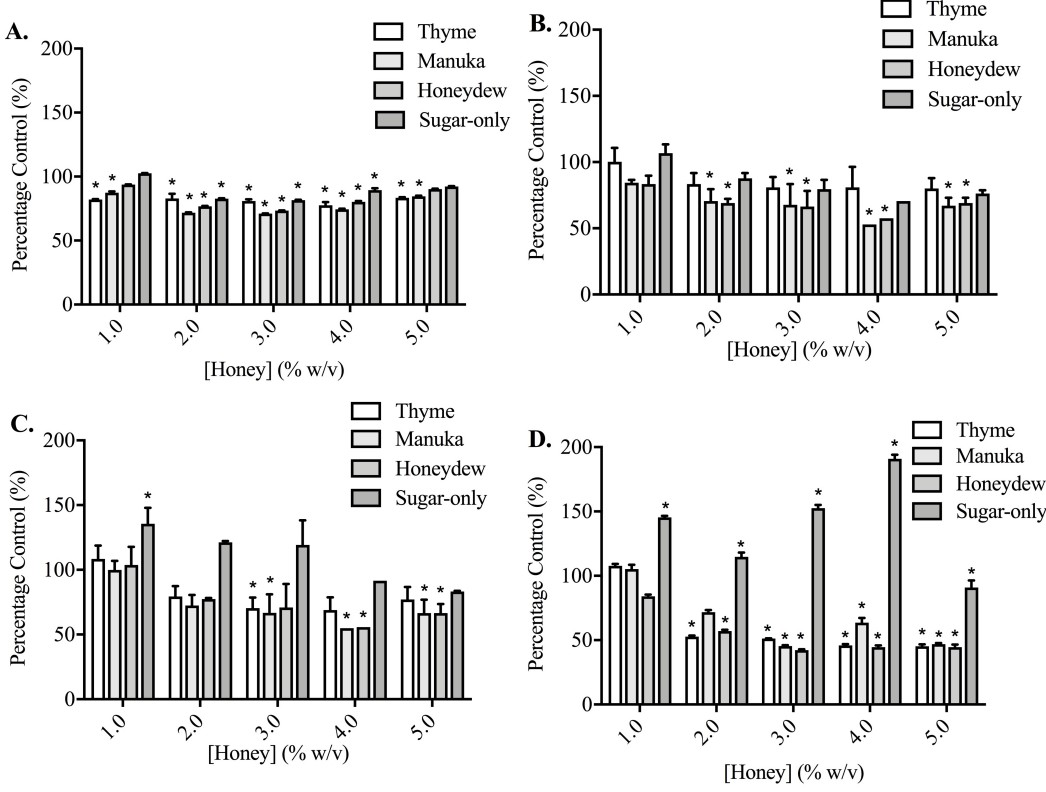

**Figure 4** **Effect of honey on PC3 and DU145 cell adhesion to fibronectin.** (A,B) PC3 and (C,D) DU145 cells were left to adhere to plates coated with fibronectin for (A) 30, (B,C) 60 or (D) 90 min. Experiments were completed in triplicate, with results expressed as mean percentage of control $\pm$ S.E.M ($n = 3$). Individual data were analysed using a two-way ANOVA followed by a Bonferroni post-hoc test, where $p < 0.05$ was required for a statistically significant difference. * represents a significant difference between vehicle-treated control (0% w/v RPMI only) and individual treatment.

Adhesion measurements for both cell lines to collagen I and fibronectin were used to determine whether the five honey-derived phenolic compounds were responsible for the loss of cell attachment seen from honey administration (Fig. 5). At 60 min, no compound had any effect on either PC3 or DU145 cell adhesion to fibronectin at any concentration. Only quercetin and kaempferol (150 μM) caused any significant change to cell adhesion to collagen I. Quercetin (150 μM) caused a reduction in PC3 cell adhesion to collagen I of 47.14 $\pm$ 10.27%. Kaempferol (150 μM) caused an increase of PC3 and DU145 cell adhesion to collagen I of 117.20 $\pm$ 11.10% and 104.10 $\pm$ 10.09%, respectively.

## DISCUSSION

We have shown that honey has anti-metastatic activity in two prostate cancer cell lines, in an *in vitro* situation in which their unmetabolised forms are in contact with the cancer cells. We have also examined some of the constituents of honey, including the major sugars and five of the main phenolic compounds, and have shown that they can partially inhibit some metastatic properties as well. Whole honeys, as well as sugars, to a lesser extent,

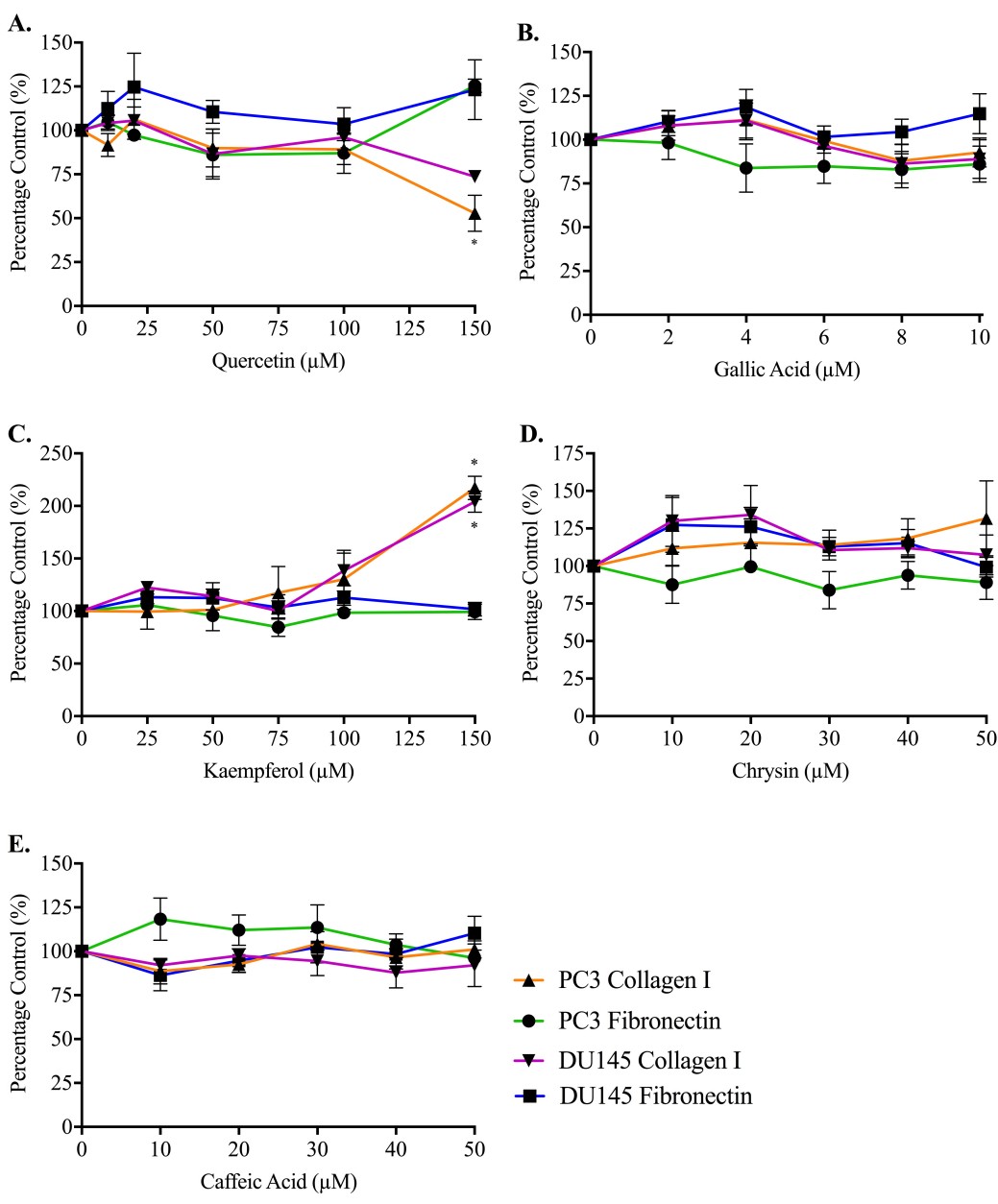

**Figure 5** **Effect of honey-derived phenolic compounds on PC3 and DU145 cell adhesion.** PC3 and DU145 cells were treated with (A) quercetin (0–150 μM), (B) gallic acid (0–10 μM), (C) kaempferol (0–150 μM), (D) chrysin (0–50 μM) or (E) caffeic acid (0–50 μM) for 60 min. Experiments were completed in triplicate, and values were expressed as mean percentage cell viability ± S.E.M ($n = 2$ or 3). Data were analysed using a two-way ANOVA followed by a Bonferroni post-hoc test, where $p < 0.05$ was required for a statistically significant difference. * represents a significant difference between control and individual treatment.

could prevent cell adhesion to collagen I, although phenolic compounds were not effective. Honeys and quercetin also strongly reduced invasion, but the effect on migration was much smaller.

We began by measuring the concentrations of five phenolic compounds in whole honeys, chosen due to their abundance and reported biological activity. An enhanced HPLC gradient system was developed which improved upon previously reported methodologies used with honey (*Yao et al., 2003*) and allowed a more sensitive detection and accurate calculation of the phenolic concentrations. For example, quercetin was found in thyme and manuka honeys, whereas using other methods, it has previously been undetectable in thyme honey (*El-Hady & Shaker, 2013*).

The phenolic compounds were measured as both free and total phenols. Free phenolic extract samples contained the unbound phenolic compounds (aglycones) within the honey, which were readily extracted by ethyl acetate. Total phenolic extract samples contained both phenolic aglycones and bound phenolic glycosides which required the hydrolysis of the glycosidic ester bonds to release phenolic aglycones before they could be extracted into an organic solvent (*Wahdan, 1998*).

A higher recovery of gallic acid and caffeic acid were observed in total phenol than in free phenol extracts, which were in contrast to the flavonoid compounds that showed a higher recovery in free phenol extracts. An explanation for this may be that flavonoids in honey exist mainly as aglycones and were detected in the free phenol samples fully, however, the two phenolic acids caffeic acid and gallic acid exist predominantly as glycosides from pollen, and therefore may require hydrolysis (as used for the 'total' samples) (*Ferreres et al., 1991*). The only exception was the gallic acid in manuka honey, of which a greater amount was detected in the free phenol extracts. This could be explained by the fact that manuka honey has been reported to have a high concentration of glucose oxidase which can produce hydrogen peroxide (*White & Doner, 1980*), and radicals including hydrogen peroxide are able to hydrolyse the glycosidic bonds between phenolics and sugars (*Brudzynski et al., 2011*; *Hussein et al., 2011*). As we found here, gallic acid was previously described as the predominant phenolic acid in both Australian and New Zealand honeys (*Yao et al., 2003*; *Yao et al., 2004*).

The effect of both whole honey and phenolic compounds on cell migration in Boyden chambers was limited but significant. The effect was found with both thyme and honeydew honeys, but the reductions were not as large as those caused by kaempferol and chrysin, of up to 50%. The sugar-only mixture, manuka honey and other phenolics did not affect migration. In contrast, *Ho et al. (2010)* reported that gallic acid (3.5 µM) could inhibit the migration of gastric cancer AGS cells by 60% in Boyden chambers, but this was measured after only 6 h of migration following 48 h of exposure to gallic acid in other wells, quite a different set-up from that used here.

Interestingly, the effect of honeys on invasion, the movement through Matrigel as well as the Boyden chamber, was much more pronounced, with honeydew honey having the greatest effect in reducing invasion by around 75%, but with all honeys having a large significant effect. Of the phenolic compounds, only quercetin and caffeic acid showed

statistically significant reductions, but there was a trend towards reduction for chrysin, gallic acid and the sugar-only mixture as well.

This demonstrates that honeys can inhibit the invasion process more strongly than the migration process, and suggests that a major point of inhibition is likely to be a part of a pathway that is more important for invasion than migration. Although we have not investigated this here, the mechanism of action may be through inhibition of the MMPs, which are among the proteases expressed at the leading edge of metastasising cells, where they facilitate the breakdown of the extracellular matrix (*Friedl & Wolf, 2003*). *Ho et al. (2010)* showed that gallic acid could reduce gelatinolytic activity of MMP-2 and MMP-9, possibly via NF-κB. It has been shown that honey can reduce the expression and nuclear translocation of NF-κB both *in vivo* and *in vitro* (*Batumalaie et al., 2013*; *Hussein et al., 2013*), although honeys were also shown not to inhibit NF-κB activity in glioblastoma cells U87MG (*Moskwa et al., 2014*). However, *Moskwa et al. (2014)* did show that honeys could reduce enzymatic activity of MMP-2 and MMP-9. Fir honey also inhibited migration of human keratinocytes through the reduced expression of MMP-9 (*Majtan et al., 2013*). *Lee et al. (2004)* demonstrated that 20 μM quercetin or luteolin could inhibit the secretion of MMPs in the MIA PaCa-2 cell line. Quercetin (50–100 μM) has also been reported to downregulate the expression of both MMP-2 and -9 in PC3 cells (*Vijayababu et al., 2006*).

Phenolic compounds have also been previously shown to affect other parts of the cell migration process, including both Rho family GTPases and integrin expression, although this has not been demonstrated in cancer cells. For example, gallic acid (25–100 μM) inhibited RhoA protein expression and activity in scar-derived fibroblasts after TGFβ stimulation (*Hsieh et al., 2016*). In L929 fibroblasts, following treatment with 20 μM quercetin for 24 h, $\alpha_V$ integrin was upregulated by 18%, and β1 integrin was similarly downregulated (*Doersch & Newell-Rogers, 2017*). Some of these concentrations were much higher than those used in our study, however there may have been a cumulative effect of the combination of lower concentrations of many phenolic compounds.

Another necessary feature of the migration and invasion processes is adhesion to the extracellular matrix. For the first time, it was shown that over the short 30–90 min timespan of the adhesion assay using collagen I, honey caused a loss of cell adhesion of more than 90% in both cell lines, although the DU145 cell line was more sensitive. The sugar constituents in honey do appear to play a role in this loss of adhesion, but do not account for the full effect. When fibronectin was used instead of collagen I, the reduction in adhesion was greatly reduced in the PC3 cells. In the DU145 cells, a decrease in adhesion of up to 50% was still found, but the sugar-only mixture increased, rather than decreased, adhesion, suggesting that the mechanism of action of the sugars is very much dependent upon the protein substrate. The differences in adhesion changes between the two cell types may relate to their expression of integrins. DU145 cells have been shown to highly express $\alpha_v$ and β1 integrins, which preferentially bind to fibronectin, and may reduce migration and result in a less invasive phenotype compared to PC3 cells (*Ruoslahti & Giancotti, 1989*; *Witkowski et al., 1993*). The increased invasiveness of PC3 cells compared to DU145 cells has instead been attributed to the expression of $\alpha6$ and β1 integrins, which bind better to collagen and laminin (*Witkowski et al., 1993*; *Suyin, Holloway & Dickinson, 2013*). Collagens, including

collagen I, are deposited at a higher rate in tumours, often in a linearised manner, which increases stromal stiffness and contributes to cancer cell migration (*Zhu et al., 1995*; *Egeblad, Rasch & Weaver, 2010*). Thus, the possibility of reducing the adhesion of prostate cells to collagen would, if translatable *in vivo*, be of potential therapeutic benefit.

By contrast, phenolic compounds do not appear to play a role in reducing prostate cancer cell adhesion. The highest concentration of quercetin (150 μM) only lowered adhesion of the PC3 cells attaching to collagen I, and 150 μM kaempferol increased adhesion for both cell lines on collagen I. Quercetin may affect adhesion through the Epidermal Growth Factor Receptor, which mediates DU145 cell adhesion to collagen I (*Lamb, Zarif & Miranti, 2011*) and can be downregulated by quercetin (*Kumar et al., 2008*; *Bhat et al., 2014*). We note that to observe biological activity in our assays with these compounds, much higher (μM) concentrations of the phenolic compounds were required than are found within the honey and therefore any effects are very unlikely to be due to a single phenolic compound.

The lack of effect of phenolic compounds on adhesion of prostate cancer cells was unexpected, given the role of these compounds in the adhesion process in other cell types. There may be some cell-type or integrin-subtype specificity. For example, chrysin (3 μM and 10 μM) was shown to inhibit collagen-induced platelet aggregation, by reducing P-selectin and integrin $\alpha$IIb$\beta$3 signaling (*Liu et al., 2016*), and caffeic acid at up to 100 μM could also lower platelet aggregation, through the same two mechanisms (*Lu et al., 2015*). Caffeic acid at 1–20 μM also lowered adherence of monocytes to human umbilical vein endothelial cells, through suppression of six different adhesion molecules and integrins (*Lee et al., 2012*).

It is often assumed that phenolic compounds are responsible for the biological activity of many natural products, including honeys. As honey contains multiple polyphenols, the effect of combination treatment *in vivo* and *in vitro* is of interest. We found that phenolic compounds were present in honey in the nM range (Table 1), however when individually used *in vitro* at μM ranges, they demonstrated minimal anti-metastatic properties. This suggests that the benefit of honey may be due to a combination of many compounds, and not individual activities.

Inhibition of cancer cell adhesion by whole honey has not previously been reported. *Maddocks et al. (2013)* reported that manuka honey (16–50% w/v) could inhibit the adhesion of 8 bacteria strains to fibronectin, fibrinogen and collagen. This was thought to be due to a reduction in fibronectin binding proteins, as well as the inhibition of biofilm production (*Maddocks et al., 2013*; *Maddocks et al., 2012*). In metastasis, the cell must be able to regulate its attachment to the surroundings in order to move forward. The level of disruption that treatment with honey causes to this process would leave metastasis unable to proceed (*Bendas & Borsig, 2012*).

We have also made the novel finding that the sugar components of honeys, as well as the phenolics, play a role in its *in vitro* inhibition of cancer adhesion, migration and invasion. Sugars may act as antioxidants in the body, in a similar way to that reported for polyphenols. A single oral administration of honey (1.5 g/kg) in humans was shown to increase the total phenolic, antioxidant and reducing capacity of the plasma. The sugar-only control, corn syrup, did not increase the total plasma phenolic levels, however, it did significantly increase

plasma antioxidant capacity (*Schramm et al., 2003*), most likely through the formation of Maillard products or by acting as reducing sugars (*White & Doner, 1980*; *Maillard, 1912*). This suggests that the activity of honey is due to presence of both phenolics and sugars, and may enhance its overall biological activity compared to other natural phenol sources lacking in sugar.

## CONCLUSIONS

We have shown that honeys and some of their constituents are able to inhibit pro-metastatic properties including migration and invasion in prostate cancer cell lines. This is likely to be related to a blocking of the adhesion process, which has been shown for the first time to be particularly strongly downregulated by honeys, and is a process that contributes to both migration and invasion. Further investigation of the mechanisms of action of honey compound combination effects in metastasis are warranted.

### Funding

This work was funded by the Department of Pharmacology and Toxicology, University of Otago. The funders had no role in study design, data collection and analysis, decision to publish, or preparation of the manuscript.

### Grant Disclosures

The following grant information was disclosed by the authors:
Department of Pharmacology and Toxicology, University of Otago.

### Competing Interests

The authors declare there are no competing interests.

### Author Contributions

- Sean D.A. Abel conceived and designed the experiments, performed the experiments, analyzed the data, prepared figures and/or tables, authored or reviewed drafts of the paper, approved the final draft.
- Sumit Dadhwal conceived and designed the experiments, performed the experiments, analyzed the data, contributed reagents/materials/analysis tools, approved the final draft.
- Allan B. Gamble conceived and designed the experiments, analyzed the data, contributed reagents/materials/analysis tools, approved the final draft.
- Sarah K. Baird conceived and designed the experiments, analyzed the data, contributed reagents/materials/analysis tools, prepared figures and/or tables, authored or reviewed drafts of the paper, approved the final draft.

### Data Availability

The raw data are provided in a Supplemental File.
## Supplemental Information

Supplemental information for this article can be found online at http://dx.doi.org/10.7717/peerj.5115#supplemental-information.

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
