# Peer review of "Honey reduces the metastatic characteristics of prostate cancer cell lines by promoting a loss of adhesion"

_PeerJ, doi:10.7717/peerj.5115_

## Round 0.1 · original submission · Major Revisions

I appreciated the comparative nature of the study in terms of the analysis of phenolic compounds and the biological effects - migration, invasion and adhesion. I find it complex and well designed, but amendment is needed.

Major points to address:

1. clear emphasis on the novelty of the study as compared to the literature
2. although the aims of the study is clearly stated, and the mechanism of action was not among them, both reviewers expressed that mechanistic aspects are missing. Potential action of the phenolic compounds and their differences need to be discussed.
3. justification of the selection of the prostate cancer cell lines
4. Unfortunately several cell lines were mixed up in labs in the past and many results became questionable in the literature. There are strict guidelines related to the use of cell lines. The cell lines used in the study come from another laboratory. Were they purchased from a reliable source like ATCC? Were they tested for authenticity? What was the passage number used?

Minor:

1. Table 1. Retention time and spectra should be removed. Spectra are already given in methods. Retention time can be added to results. For concentrations numbers with one decimal digit would be enough precise.
2. Methods: Please use "Cell culture" instead of "maintenance".
3. Boyden chamber
Were these commercially available culture inserts, like Transwells from Costar? The parameters (membrane type, multiwell type) and the supplier need to be specified.
4. It is clear that the effects were compared to control groups on Fig. 2-5. Please make it more clear by harmonizing the labels on Y axes and specifying what was the vehicle in both the methods and on all figure legends. Importantly, what was the final concentration of DMSO in the treatment media?

Reviewer 1 ·

Basic reporting

Abel et al in this manuscript have attempted to demonstrate the anti-metastatic activity of various honey from N. Zealand (Thyme, Manuka and Honey Due) using prostate cancer cells PC-3 and DU-145 cells. The authors do not highlight any mechanisms to suggest their hypothesis. Reported literature already suggests honey as an anti-metastatic agent having ability to alter matrix metalloproteinases. Therefore the work performed by authors is not new to this area of investigation though their experiments have a direct approach suggesting the anti-metastatic action of honey.

Experimental design

The authors list the phenolic contents of honey. The authors have not accounted for the difference in the contents in all three honey: Thyme, Manuka and Honey Due . This might suggest the difference in the activity which may differ from the other.

Figure 1-5 have not used a vehicle control in their experiment to make a comparison.

The authors did not show the effect of honey on the proteins involved in metastasis (invasion or migration or adhesion) to effectively draw conclusions from a mechanistic point.

Validity of the findings

The authors have used two aggressive prostate cancer cell lines to prove their hypothesis. It would be ideal if they use some more metastatic cell lines along with a normal cell counterpart.

Additional comments

Additional experiments are needed to substantiate these results.

Reviewer 2 ·

Basic reporting

Background is clear

Experimental design

Research question well defined and methods as well

Validity of the findings

Data is robust and the conclusion are well stated

Additional comments

In this paper, Baird present that Honey can reduce the metastatic characteristics of prostate cancer cell lines by promoting a loss of adhesion .

The manuscript is well written. Although the manuscript contains some valuable information obtained in vitro experiments. So, the article possess a practical value. In the background of the study the purpose of the article is clearly stated, data is robust and statistically sound, the conclusion is well stated and limited to supporting results. However some specific comments are cited below;

1. Catenin is a protein, originally described as an integral part of intercellular adhesion system. Today it is considered an oncoprotein likely to activate proliferation and inhibit apoptosis. why it was not studied

2. Since honey is mixture of phenolic compounds how it’s possible to exprime the concentration in nM ? Table 1 shows the concentrations of free and total phenols found in the thyme, manuka and honeydew honeys, as both µg/100g of honey and in nM (page 11line 255 and 256)

3. Gallic acid concentration at 1082.88 ± 16.25 µg/100g is the equivalent of 10,8 mg/g it seems very low to exert any activity (page 11 line 259)
4. How Sugars can act as antioxidants in the body, in a similar way to that reported for polyphenols as reported in page 19 line 450 (It was necessary to carry out the DPPH test to evaluate the antioxidant activity of honeys).

5. The polyphenols found in honeys are the same found in propolis from the same regions. why the authors did not compare the honey and propolis activities ?

6. Indeed, The honey inhibitory activity of MMP3 and 9 is low compared to that observed with a phenolic compound of Algerian propolis (Segueni et al, 20011 Planta Medica). Chicoric acid inhibits MMP-3, MMP-7 and MMP-9 with IC 50 of 0.63, 0.10 and 2.38 ug / ml

7. why the authors did not realize western blotting to quantify the variations of the tumoral proteins like integrins, intracellular signalling proteins, fibronectin…?

---

## Round 0.2 · accepted · Accept

All the comments were adequately answered.

Please correct the sentence in the discussion at line 420 (marked up copy) or line 452 (pdf) which seems to be unfinished, during the final production phase.

#